# Osteoclast-Driven Osteogenesis, Bone Remodeling and Biomaterial Resorption: A New Profile of BMP2-CPC-Induced Alveolar Bone Regeneration

**DOI:** 10.3390/ijms232012204

**Published:** 2022-10-13

**Authors:** Hongzhou Shen, Yu Zhuang, Chenglong Zhang, Changru Zhang, Yuan Yuan, Hongbo Yu, Jiawen Si, Guofang Shen

**Affiliations:** 1Department of Oral and Craniomaxillofacial Surgery, Shanghai Ninth People’s Hospital, College of Stomatology, Shanghai Jiao Tong University School of Medicine, Shanghai 200011, China; 2Laboratory for Digitized Stomatology, Research Center for Craniofacial Anomalies, Shanghai Key Laboratory of Stomatology, Shanghai Research Institute of Stomatology, Shanghai 200011, China; 3Key Laboratory for Ultrafine Materials of Ministry of Education, East China University of Science and Technology, Shanghai 200237, China

**Keywords:** osteoclasts, alveolar bone regeneration, bone remodeling, biomaterial resorption, calcium phosphate cement

## Abstract

This bedside-to-bench study aimed to systematically investigate the value of applying BMP2-loaded calcium phosphate cement (BMP2-CPC) in the restoration of large-scale alveolar bone defects. Compared to deproteinized bovine bone (DBB), BMP2-CPC was shown to be capable of inducing a favorable pattern of bone regeneration and bone remodeling accompanied by active osteoclastogenesis and optimized biomaterial resorption when applied in reconstructive periodontally accelerated osteogenic orthodontics (PAOO) surgery. To verify the regulatory role of osteoclasts in the BMP2-CPC-induced pattern of bone regeneration, in vitro and in vivo studies were designed to elucidate the underlying mechanism. Our results revealed that osteoclasts played a multifaceted role (facilitating osteogenesis, bone remodeling and biomaterial resorption) in the BMP2-CPC-induced bone regeneration. Osteoclasts contributed to the osteogenic differentiation of mesenchymal stem cells (MSCs) by secreting calcium ions, CTHRC1 and PDGF-B. Moreover, the increased osteoclasts promoted the remodeling of new bone and BMP2-CPC resorption, leading to a harmonized replacement of biomaterials with mature bone. In conclusion, the in vitro and in vivo experimental results corresponded with the clinical results and showed the optimized properties of BMP2-CPC in activating osteoclast-driven bone regeneration and remodeling, thus indicating the highly promising prospects of BMP2-CPC as an ideal therapeutic for alveolar bone defects.

## 1. Introduction

Alveolar bone defects that are not self-repairable can seriously damage the jawbone integrity and periodontal tissue, resulting in tooth loss, stomatognathic dysfunction and harmful esthetic impacts. To date, numerous bone grafts, including autogenous bone, xenogenic bone and synthetic bone substitutes, have been developed to facilitate alveolar bone regeneration and restore jawbone integrity. Among the currently available grafts, autogenous bone is deemed the “gold standard” material due to its superior osteoinductivity and biocompatibility and predictable outcomes [1,2]. However, donor site morbidity, variable resorption, prolonged medication duration and subsequent patient discomfort have significantly limited the clinical application of autogenous bone grafts [2]. Although xenogenic bone, such as DBB, partly overcomes the drawbacks of autogenous bone, poor osteoinductivity and potential risks of disease transmission restrict its clinical application [3,4]. To confront this dilemma, synthetic bone substitutes are believed to be a promising choice for treating alveolar bone defects, while some synthetic bone substitutes have been applied in the clinic, such as β-tricalcium phosphate (β-TCP), hydroxyapatite (HA) and calcium phosphate cement (CPC) [5,6]. Compared to the over-slow resorption rate of HA and the over-fast resorption rate of β-TCP, the improved resorption rate of CPC is more suitable for guiding bone regeneration [7,8]. However, the osteoinductivity of CPC is much weaker than that of autogenous bone, which might lead to delayed bone formation or bone malformation [9]. Based on this consideration, various growth factors (bone morphogenetic protein 2, transforming growth factor-β, insulin-like growth factor I, etc.) have been introduced into synthetic bone substitutes to improve their osteoinductivity [1,10]. Among these growth factors, bone morphogenetic protein 2 (BMP2) is thought to be the most potent osteoinductive factor and has been widely reported to significantly promote the osteogenic differentiation of MSCs and new bone formation in both experimental and clinical practice [5,10]. The incorporation of BMP2 markedly enhanced the bone restorative effect of synthetic bone substitutes and greatly expanded the horizon for the clinical application of synthetic bone substitutes [1,5,6,9]. However, due to the short half-life of BMP2, the in vivo BMP2 is prone to degradation, with the result that the local effective concentration of BMP2 is hard to maintain [11,12]. To this end, controlled-release scaffolds are used to preserve the activity of BMP2. Lin et al. [13] indicated that the present BMP2-CPC biomaterial showed an initial burst release within 24 h followed by a slow sustained release. The structural analysis results demonstrated that the secondary structure of CPC-released BMP2 was well maintained, and the sustained release effectively established local accumulation of BMP2 [13].

Previously, we evaluated the bone restorative effect of BMP2-CPC in the setting of guided bone regeneration (GBR) and found that BMP2-CPC exhibited stronger osteogenic capability than DBB did [14]. BMP2-CPC was capable of initiating an improved immune response characterized by upregulated M2-phenotype polarization in local macrophages, which subsequently alleviated biomaterial-stimulated inflammatory reaction and enhanced osteogenesis and angiogenesis [15]. When applied in treating long bone fractures, BMP2-CPC also led to satisfactory clinical performance in terms of a shortened bone-repairing period and improved posttraumatic bone quality [13]. These results showed that BMP2-CPC featured excellent biocompatibility, osteoconductivity, osteoinductivity and immunoregulatory properties. Notably, in terms of large bone defect restoration, the bone regeneration process and bone remodeling process are equally important and are orchestrated by the cross-talk between osteoblasts and osteoclasts [16]. Osteoclasts were reported to promote the bone defect healing process through the secretion of pro-osteogenic factors [17,18,19], and depletion of osteoclasts was shown to be detrimental to osteogenesis [19,20]. Moreover, osteoclasts play a crucial role in remodeling newly formed bone tissue and in CPC particle resorption [2,8,21,22]. This evidence strongly indicates that a fine-tuned osteoclastogenesis process is an important prerequisite to achieve favorable bone defect restoration. However, the effect of BMP2-CPC on osteoclastogenesis and subsequent osteoclast-driven osteogenesis, bone remodeling and biomaterial resorption is still unclear.

In this bedside-to-bench study, we first report the clinical application and restorative effect of BMP2-CPC in reconstructive periodontally accelerated osteogenic orthodontics (PAOO) surgery, which is designed to facilitate large-scale alveolar bone regeneration and periodontal remodeling. Compared to DBB, a bone graft frequently used for PAOO surgery, BMP2-CPC was highly active in promoting local osteoclastogenesis, osteogenesis and bone remodeling in the grafted area. Since osteoclasts are widely reported to participate in the resorption of calcium phosphate biomaterials (CaPs) and bone remodeling processes [8,21], we further designed in vitro and in vivo studies to investigate the role of osteoclasts in BMP2-CPC-stimulated bone regeneration and remodeling. In this bedside-to-bench study, clinical and experimental results were combined to shed new light on the osteogenic mechanism of BMP2-CPC in alveolar bone restoration.

## 2. Results

### 2.1. Clinical Study

#### 2.1.1. Radiographic Evaluation

Among the enrolled patients, 18 patients with a mean age of 26.06 ± 4.85 years were included in the DBB group, and 19 patients with a mean age of 24.05 ± 4.15 years were included in the BMP2-CPC group. Uneventful healing was achieved in all patients, and no major complications were recorded in the 6-month follow-up period. Postsurgical CT images showed that the implantation of both DBB and BMP2-CPC effectively remedied the bone insufficiency in the lower anterior region, as evidenced by the significantly increased alveolar bone height and thickness values (Figure 1a). The measurement results of VBH and HBT indicated no significant postsurgical differences between the DBB and BMP2-CPC groups (Table 1), while decreasing trends in the postsurgical VBH and HBT values were observed in both groups (Figure 1b). Figure 1c demonstrates that no significant differences were detected in the reduction values of VBH and HBT between the DBB and BMP2-CPC groups. Similarly, there were no significant differences in the reduction rates of VBH and HBT between the DBB and BMP2-CPC groups (Figure 1d). The mean postsurgical reduction values and rates of VBH and HBT are listed in Table 2.

#### 2.1.2. Histological Examination

The healing patterns of the alveolar bone defects in each group were examined by HE staining, and representative images are presented in Figure 2a. The HE results showed that both DBB and BMP2-CPC were capable of facilitating bone regeneration at the defect sites, but the healing patterns related to the two biomaterials were significantly different. The grafted DBB particles were encapsulated by fibrous tissue that housed scattered newly formed bone. No visible phenomena associated with osteoclastogenesis were observed in the DBB group. Notably, BMP2-CPC stimulated an improved healing pattern characterized by robust bone regeneration and mature bone structure, while vascularized bone marrow-like tissues and multinucleated osteoclast cells were found in the grafted area. Moreover, larger blue-stained areas were observed in the DBB group, which demonstrated that the new bone formation around the DBB was immature (Figure 2b). In contrast, the new bone of the BMP2-CPC group was more mature and accompanied by enhanced expression of the osteogenic marker OCN (Figure 2c). The TRAP staining results further supported the HE finding that BMP2-CPC promoted local osteoclastogenesis in the grafted area, as evidenced by the increased number of red multinucleated osteoclasts in the BMP2-CPC group (Figure 2d). Quantitative analysis indicated that BMP2-CPC stimulated significantly more new bone formation than DBB (Figure 2e), and fewer BMP2-CPC particles remained in the defects in the BMP2-CPC group (Figure 2f).

### 2.2. In Vitro Study

#### 2.2.1. Biomaterial Microstructure

As depicted in Figure 3a, an interconnected porous structure was observed in the CPC particles. Extensive needle-like hydroxyapatite crystals were found on the surface of CPC particles under a higher magnification. The micropores of CPC particles formed perfect sites for protein adsorption, and the entrapped BMP2 protein was uniformly distributed on the surface of micropores.

#### 2.2.2. Cell Viability and Differentiation of Bone Marrow Macrophages (BMMs)

As shown in Figure 3b, BMMs attached to the surfaces of both CPC and BMP2-CPC exhibited good viability with no obvious sign of cell death. Figure 3c shows that both CPC and BMP2-CPC had no significant influence on the proliferation of BMMs. As the precursor cells of osteoclasts, undifferentiated BMMs manifested as round mononucleated cells on day 1. With the stimulation of M-CSF and RANKL, BMMs attached to the surfaces of both CPC and BMP2-CPC successfully differentiated into multinucleated osteoclasts on day 7, whereas those treated with M-CSF alone exhibited polygonal or elongated spindle shapes with no sign of osteoclast differentiation (Figure 3e).

#### 2.2.3. Osteoclast Activity

Figure 3f shows that the TRAP activity was significantly higher in cells treated with M-CSF and RANKL than in those treated with M-CSF alone. This phenomenon was in line with the aforementioned finding of osteoclast differentiation, and, more importantly, the TRAP activity was further elevated in the cells stimulated with BMP2-CPC in the osteoclast-differentiation groups. Following the enhanced osteoclast differentiation in the group treated with BMP2-CPC, M-CSF and RANKL, the resorption rate of CPC particles was accelerated accordingly, as evidenced by the significantly increased Ca^2+^ concentration in the culture medium containing BMP2-CPC, M-CSF and RANKL (Figure 3g).

Analysis of osteogenic factors indicated that the expression of *CTHRC1* and *PDGF-BB* was significantly upregulated in response to stimulation with M-CSF and RANKL. BMP2-CPC upregulated the expression of *CTHRC1* in osteoclast-commitment cells to a greater extent than CPC. Regarding other osteoclast-secreted osteogenic factors, *Wnt10b* and *BMP6* expression levels did not significantly differ among the four groups (Figure 3h). Further ELISA results indicated that PDGF-BB secretion was significantly increased in the osteoclast-commitment cells stimulated with BMP2-CPC (Figure 3d).

#### 2.2.4. Osteogenic Differentiation of MSCs

CCK-8 analysis revealed that the MSC viability was normal in all groups and that the CM from each group had no significant influence on the proliferation of the MSCs (Figure 4a). Figure 4c shows the ALP staining results of MSCs treated with various CMs for 5 or 7 days. More positive staining was observed in the CM2, CM3 and CM4 groups at day 5, and ALP-positive staining areas were further increased in these groups at day 7. Quantitative analysis revealed significant upregulation of ALP activity in the MSCs treated with CM2 and CM4. In particular, the activity of ALP in MSCs treated with CM4 was stronger than that in cells treated with CM2 (Figure 4e). Similarly, positive ARS staining was observed in the MSCs treated with CM4 as early as day 14. CM2 was not capable of inducing the mineralization of MSCs until day 21 (Figure 4d). Quantitative analysis of calcium deposition indicated that CM2 and CM4 significantly promoted the late-stage osteogenic differentiation of MSCs compared with that achieved with CM1 and CM3. CM4 exhibited a strong ability to facilitate the mineralization of MSCs, whereas the ability of CM2 was weaker (Figure 4f).

The immunofluorescence staining results revealed that COL-I expression was significantly enhanced in the CM4 group compared to the other groups (Figure 4b). Western blotting revealed that OCN expression was significantly upregulated in the MSCs treated with CM2 and CM4. CM4 led to higher OPN and OCN production than CM2 (Figure 4g,h).

### 2.3. In Vivo Study

#### 2.3.1. Micro-CT Analysis

The reconstructed images of rabbit calvarial defects showed good postsurgical healing and obvious bone regeneration in the CPC and BMP2-CPC groups, whereas a limited amount of new bone accumulated on the margin of the defects in the Con group (Figure 5a). Transverse images showed that the healing patterns of the calvarial defects filled with CPC or BMP2-CPC were characterized by a combination of biomaterial particles and newly formed bone. In contrast to the chaotic bone structure of the CPC group, the bone structure of the BMP2-CPC group seemed well organized at 3 months after surgery, as evidenced by the two-layer cortical bone-like structure in the BMP2-CPC group (Figure 5b). Quantitative analysis of new bone volume and density demonstrated that both CPC and BMP2-CPC promoted obvious bone regeneration in the calvarial defects; however, no significant differences in these two bone parameters were found between the CPC and BMP2-CPC groups (Figure 5c,d).

#### 2.3.2. Histological Examination (Undecalcified)

Figure 5e shows that the defects of the Con group were filled with fibrous tissues with no visible sign of bone regeneration. In the CPC and BMP2-CPC groups, the healing patterns of calvarial defects appeared as a combination of new bone and biomaterial particles encapsulated by fibrous tissues, which agreed with the findings of the micro-CT examination. The direct bone-to-biomaterial contact indicated good biocompatibility of CPC and BMP2-CPC. Quantification of the new bone area revealed significant bone regeneration in the defects treated with CPC and BMP2-CPC, and BMP2-CPC induced more new bone than CPC (Figure 5f). Notably, compared to the large quantities of residual CPC particles, fewer BMP2-CPC particles were observed in the calvarial defects, suggesting that the resorption of BMP2-CPC particles was accelerated (Figure 5e,g).

#### 2.3.3. Histological and Immunohistochemical Examination (Decalcified)

As shown in Figure 5h, the 10-mm-diameter calvarial defects could not be self-healed by the rabbits. Thus, the defects in the Con group healed via a fibrous connection, and no obvious signs of bone regeneration were observed at 1.5 months after surgery. Sporadic new bone was observed in the margin of the defects at 3 months after surgery, and most of the defect areas were still filled with fibrous tissues. Compared to the worse osteogenesis in the untreated group, both CPC and BMP2-CPC facilitated local bone regeneration to varying degrees. CPC boosted a weaker bone regeneration pattern characterized by a limited amount of new bone that progressed from the residual bone margins. The CPC particles and new bone were encapsulated by fibrous tissues that housed numerous inflammatory cells. Although new bone was increased in the CPC-filled defects at 3 months after surgery, the local inflammatory reaction was still apparent, and the bone structure was chaotic. Notably, the bone regeneration of the BMP2-CPC group manifested as an osteoinductive pattern. The scattered BMP2-CPC particles provided a number of growth centers for new bone formation, and the accumulated new bone subsequently merged into a bone bridge to fill in the defects. With the remodeling of new bone, the bone structure of the BMP2-CPC group tended to be mature at 3 months after surgery. Vascularized bone marrow tissues were distributed in the trabecular bone embedded in a two-layer cortical bone-like structure. Quantitative analysis of the new bone area indicated significant bone regeneration in the CPC and BMP2-CPC groups and that BMP2-CPC induced a stronger osteogenesis pattern than CPC (Figure 5i). Accordingly, the expression of the osteogenesis-related markers, OCN and OPN, was upregulated in the BMP2-CPC group as early as 1.5 months after surgery, while the expression of only OCN was increased at 1.5 months after surgery in the CPC group. The expression levels of OPN and OCN were further elevated in the BMP2-CPC group at 3 months after surgery, and BMP2-CPC induced the production of more OPN and OCN in the local new bone tissues than CPC (Figure 6a,b).

#### 2.3.4. Osteoclast Differentiation

The in vivo differentiation of osteoclasts in local tissues was investigated by TRAP staining, and representative images are presented in Figure 6c,e. Consistent with the clinical histological results, osteoclast differentiation was significantly enhanced in the BMP2-CPC-filled defects. Positive staining sites were distributed around BMP2-CPC particles and newly formed bone, in which multinucleated osteoclasts were found under a high magnification. Similarly, CPC stimulated a weaker pattern of osteoclast differentiation than BMP2-CPC. Compared to the CPC and BMP2-CPC groups, no obvious phenomenon associated with osteoclast differentiation was observed in the Con group. Quantitative analysis indicated that although both CPC and BMP2-CPC were capable of promoting osteoclast differentiation, BMP2-CPC exhibited an enhanced ability to induce local osteoclastogenesis (Figure 6d).

## 3. Discussion

CPC has been clinically used in the treatment of bone defects and fractures since its first emergence in the 1980s. Despite the favorable biocompatibility of CPC, its osteoinductivity and biodegradability are still far from satisfactory [7,23]. Among the extensive means to improve the biological properties of CPC, we fabricated a precured BMP2-CPC microscaffold and proved that the incorporation of BMP2 significantly increased the bone regenerative performance of CPC in repairing rabbit distal femur defects and in the clinical treatment of long bone fractures [13]. Our previous study showed that BMP2-CPC significantly enhanced the in vivo bone regeneration and remodeling of rabbit calvarial bone defects, providing preliminary support for its clinical application [14,15]. However, the effect of BMP2-CPC on the clinical treatment of larger-scale alveolar bone restoration is still unclear.

In the current bedside-to-bench study, we first report the clinical application and excellent restorative effect of BMP2-CPC in PAOO surgery. According to the radiographic evaluation results, BMP2-CPC was as good as DBB in remedying vertical and horizontal alveolar bone insufficiency. The shrinking patterns of the postsurgical bony contour were also consistent between the DBB and BMP2-CPC groups. Quantitative analyses of the mild reduction values and rates of VBH and HBT indicated that BMP2-CPC and DBB had similar abilities to preserve the stability of bony contours post surgery. However, the findings derived from the clinical radiographic examination should be interpreted with caution due to the similar inorganic compositions of DBB and BMP2-CPC to those of bone tissues that might interfere with the identification of bone area in CT images [24]. Therefore, histological examinations were performed to further reveal the bone regeneration pattern induced by DBB or BMP2-CPC. Notably, postsurgical histological examinations of selected patients in this study demonstrated that BMP2-CPC induced more robust bone regeneration, characterized by significantly increased newly formed bone and orderly arranged bone structure, whereas DBB induced only limited amounts of new bone and immature bone structures. These clinical histological findings were highly consistent with our previous in vivo experimental results [14], providing direct evidence for the application of BMP2-CPC in the restoration of alveolar bone defects. Interestingly, the improved bone regeneration pattern induced by BMP2-CPC was accompanied by enhanced local osteoclast formation. Unlike the traditional perception, the narrow description as a bone destroyer does not fit the multifaceted roles of osteoclasts in the current perspective. According to previous studies, osteoclasts play a pivotal role in the remodeling of newly formed bone tissue and in CPC particle resorption [2,8,21,22]. Moreover, osteoclasts were reported to be involved in the CaP-stimulated osteogenic process [17,18,19], and depletion of osteoclasts was shown to be detrimental to osteogenesis [19,20]. This evidence strongly indicates that osteoclasts may orchestrate the fine-tuned kinetics of osteogenesis, bone remodeling and BMP2-CPC resorption in the healing period.

Enlightened by our clinical histological findings, we concentrated on exploring the regulatory role of osteoclasts in BMP2-CPC-stimulated bone regeneration and remodeling through in vitro and in vivo studies. Osteoclasts, which originate from hematopoietic precursor cells, require M-CSF and RANKL for their survival, proliferation, differentiation and activation [25]. In particular, the binding of RANKL to its receptor RANK is the key factor determining the differentiation of precursor cells into osteoclasts [25,26], and the absence of RANKL here completely disrupted the osteoclast formation in the group treated with only M-CSF (Appendix A Appendix A). In this study, BMP2-CPC significantly enhanced not only the in vitro osteoclast formation induced by M-CSF and RANKL but also the in vivo bone regeneration and osteoclastogenesis in rabbit calvarial bone defects, which was consistent with the study of Itoh et al. [27]. Kim et al. [28] and Granholm et al. [29] also found that BMP2 indirectly promoted osteoclast differentiation by upregulating RANKL expression and downregulating osteoprotegerin (OPG) expression in MSCs. Moreover, previous studies reported that BMP2 released from scaffolds promoted osteoclastogenesis and osteoclast maturation as well as strengthened the osteoclast-induced osteogenic differentiation of MSCs [11,30]. Notably, our study also showed that the CM derived from osteoclasts stimulated with BMP2-CPC markedly facilitated the osteogenic differentiation of MSCs in vitro, which was supported by the fact that the submicrostructured CaP induced the osteogenic differentiation of MSCs in vitro and bone formation in vivo by upregulating osteoclast differentiation and subsequently enhancing the secretion of osteogenic factors from osteoclasts [31]. When osteoclast differentiation was inhibited in vivo, CaP-induced bone regeneration was also abrogated [20]. To further study the pro-osteogenic role of osteoclasts in BMP2-CPC-induced bone regeneration, several osteogenic factors that might participate in the osteogenic differentiation of MSCs were identified by in vitro experiments. We found that the Ca^2+^ concentration in the culture medium and the expression of *CTHRC1* and *PDGF-BB* were significantly increased in response to BMP2-CPC. According to previous studies, the increased calcium ion level in CM derived from osteoclasts stimulated with BMP2-CPC could enhance the proliferation, migration and osteogenic differentiation of MSCs by activating the calcium-sensing receptor [32]. CTHRC1, a soluble protein secreted from mature osteoclasts, was proven to target Wnt-activated inhibitory factor 1 in stromal cells and activate the PKCδ-ERK signaling pathway to promote osteogenesis [33]. PDGF-BB promoted vascularized bone regeneration in vivo by enhancing the osteogenic differentiation of MSCs and angiogenesis of vascular endothelial cells by activating the ERK1/2 and PI3K/AKT signaling pathways [34]. Moreover, CTHRC1 and PDGF-BB also functioned as guide molecules for targeting MSCs to bone defect sites, thus initiating bone-forming activity [35,36]. Together, these results indicate that BMP2-CPC-induced osteoclasts may play an important osteogenic role in the bone defect healing process by secreting calcium ions, CTHRC1 and PDGF-BB.

Besides the pro-osteogenic role, the roles of osteoclasts in bone remodeling and biomaterial resorption are equally important. With the accumulation of newly formed bone at defect sites, bone remodeling mediated by tight osteoblast–osteoclast interaction constantly occurs to reshape the bone structure into a mature structure [16]. The increased numbers of osteoblasts and osteoclasts induced by BMP2-CPC may reciprocally regulate cellular behavior, survival and differentiation through cell-to-cell contact and secretory proteins, triggering the accelerated bone maturation process in the BMP2-CPC-treated calvarial defects [13,16,35]. As a result, we observed that the BMP2-CPC-induced new bone exhibited a well-organized bone structure, whereas the bone structure of the CPC group appeared chaotic. Similarly, Seeherman et al. [37] reported that the rabbit mid-diaphyseal defects treated with 0.166 mg/mL BMP2-loaded CPC had completely regenerated cortical bone and bone marrow bridging of the defect at 8 weeks after surgery. Compared to DBB, BMP2-CPC enhanced the structural maturity of the bone, which exhibited thick trabeculae and small medullary spaces covered by a two-layer cortical bone-like structure at 3 months after surgery [14]. Regarding biomaterial resorption, osteoclasts were reported to participate in the bulk digestion of CaP particles by secreting acid [8]. In the present study, the significantly increased calcium ion concentration in the CM derived from the osteoclasts stimulated with BMP2-CPC indicated higher acid resorption of CPC [38], which was in accordance with the accelerated in vivo resorption of CPC and enhanced osteoclast activity in the BMP2-CPC group. The accelerated in vivo resorption of BMP2-CPC mediated by enhanced local osteoclast differentiation was repeated in another rabbit mid-diaphyseal defect model treated with BMP2-loaded CPC, in which Seeherman et al. [37] observed a similar phenomenon associated with enhanced osteoclast formation and accelerated CPC resorption. Likewise, Niu et al. [30] found that BMP2 accelerated bioglass scaffold resorption by enhancing preosteoclast differentiation and maturation through a synergistic mechanism involving the direct BMP2 stimulation and the indirect activation of factors secreted from MSCs. In this study, the improved resorption rate of BMP2-CPC was sufficiently correlated with the accelerated rate of BMP2-CPC-induced bone regeneration, leading to a harmonized replacement of biomaterials with new bone at defect sites. With the communication of osteoclasts and osteoblasts, bone regeneration, bone remodeling and biomaterial resorption actively occur in the BMP2-CPC-treated defects to reshape the newly formed bone to meet structural and metabolic demands, which further supports the application of BMP2-CPC in the restoration of alveolar bone defects [39].

Admittedly, although this study offers new insight into the regulatory role of osteoclasts in BMP2-CPC bone regeneration, several points merit further investigation to gain a deep understanding. First, the signaling pathway and mechanism underlying BMP2-CPC-facilitated osteoclastogenesis need to be explored. Second, the effects of the potential osteogenic factors identified by this study, calcium ion, CTHRC1 and PDGF-BB, require further study in vitro and in vivo. Third, considering the reciprocal communication between osteoclasts and osteoblasts, the influence of osteoblast lineage cells on BMP2-CPC-induced osteoclastogenesis and subsequent bone homeostasis likewise need attention. Fourth, the strength of the supporting evidence would be bolstered if the regulatory role of osteoclasts in BMP2-CPC-induced bone regeneration was elucidated in specific osteoclast-depleted mouse models. Finally, a randomized controlled clinical trial with a larger sample size and long-term observation period should be performed to improve our understanding of the advantages of applying BMP2-CPC in the restoration of alveolar bone defects in the future.

## 4. Materials and Methods

### 4.1. Clinical Study

#### 4.1.1. Study Design

This study was conducted in accordance with the principles of the Declaration of Helsinki (2013) and approved by the Independent Ethics Committee of Shanghai Ninth People’s Hospital affiliated with Shanghai JiaoTong University, School of Medicine (SH9H-2019-T217-1). The clinical application of BMP2-CPC (Rebone™, Shanghai, China) was approved by the China Food and Drug Administration (Certified No. (2013): 34-60199). A review of patient charts and radiographs was performed for all patients who underwent PAOO surgery at the Department of Oral and Craniomaxillofacial Surgery, Shanghai Ninth People’s Hospital from January 2017 to October 2019, and 37 patients (16 males and 21 females) were finally enrolled in this study.

The inclusion criteria were as follows: (1) patient was 18–40 years of age; (2) anterior mandibular region surgical site; (3) thin or incomplete labial alveolar bone (including alveolar bone dehiscence and fenestration); (4) healthy periodontal condition; (5) patient was receiving orthodontic treatment; and (6) all surgeries were performed by the same senior maxillofacial surgeon. The exclusion criteria were as follows: (1) incomplete clinical or radiographic data; (2) a history of periodontal treatment; (3) history of diabetes mellitus; (4) history of administering bisphosphonates; (5) history of alcohol abuse or smoking; and (6) missing teeth in the anterior mandibular region. The enrolled patients were divided into two groups according to the bone graft used during surgery. Patients in the DBB group underwent grafting with DBB (Bio-Oss^®^, Geistlich Biomaterials, Wolhusen, Switzerland), and patients in the BMP2-CPC group underwent grafting with BMP2-CPC.

#### 4.1.2. Surgical Procedures

After local anesthetization with Primacaine Adrenaline^®^ (Acteon Pierre Rolland, Mérignac, France), a full-thickness mucoperiosteal flap was elevated in the anterior mandibular region using a sulcular incision. After exposing the surgical field, corticotomy was performed using a piezoelectric surgical device (Piezosurgery, Silfragrant, Rome, Italy). Vertical cortical bone incisions were made in the interradicular spaces at a depth of 2 to 3 mm and were connected by a horizontal bone incision located 5 mm beneath the apices of roots. DBB or BMP2-CPC was placed on the prepared cortical bone and covered with an absorbable membrane (Bio-Gide^®^, Geistlich Biomaterials, Switzerland). The mucoperiosteal flap was coronally repositioned in a tension-free manner to cover the grafted bone grafts and membrane. Then, the wound was closed using absorbable 5-0 sutures (Vicryl, Ethicon, NJ, USA) (Figure 7a). Cephalosporin antibiotics combined with metronidazole were routinely prescribed for at least 3 days, and 0.12% chlorhexidine gluconate oral rinses were used twice daily for plaque control.

#### 4.1.3. Radiographic Evaluation

Routine CT examinations were performed at 2 weeks, 3 months and 6 months after surgery. The obtained CT data were processed using Simplant Pro Software (Materialise, Leuven, Belgium) for the measurement of parameters defined according to previous studies [40,41] (Figure 7b). The vertical bone height (VBH) was determined by measuring the distance between the labial alveolar crest and the root apex, parallel to the long axis of the lower anterior tooth. The horizontal bone thickness (HBT) was determined by measuring the alveolar bone width at the level of the root apex, perpendicular to the long axis of the lower anterior tooth. The parameters were analyzed in terms of their reduction values and reduction rates, as illustrated in Table 3.

#### 4.1.4. Histological Examination

Of the 37 enrolled patients, 3 patients in the BMP2-CPC group and 3 patients in the DBB group underwent genioplasty with bone tissue histological examination at 12 months after PAOO surgery. All participants signed an informed consent agreement for the histological examination. In detail, redundant bone slices obtained from the anterior mandibular region during genioplasty were collected for histological examination. After fixation in 4% paraformaldehyde (PFA), the harvested bone slices were decalcified in 10% ethylene diamine tetraacetic acid (EDTA), dehydrated in a gradual series of ethanol (70–100%) and embedded in paraffin for sectioning. Sections 5-μm thick were subjected to hematoxylin-eosin (HE), Masson, immunohistochemical and TRAP staining. Details of the HE, immunohistochemical and TRAP staining are provided in Section 4.3.5 and Section 4.3.6. The percentage of new bone area and residual bone grafts within the total defect area was measured in five randomly selected fields of vision using ImageJ (National Institutes of Health, USA). Osteocalcin (OCN) was labeled for immunohistochemical staining.

Masson staining: Following deparaffinization and rehydration, the prepared sections were sequentially stained with Weiger’s hematoxylin and Masson Ponceaux acid fusion. Then, the sections were immersed in 2% glacial acetic acid aqueous solution and differentiated with a 1% phosphomolybdic acid aqueous solution. Finally, the sections were transparentized and sealed, and images were captured using an upright microscope (Nikon, Japan).

### 4.2. In Vitro Study

#### 4.2.1. Preparation and Characterization of Biomaterials

The biomaterials (CPC and BMP2-CPC) used in this study were purchased from Rebone Biomaterials Co., Ltd. (Shanghai, China). CPC is mainly composed of tetracalcium phosphate and dicalcium phosphate anhydrous. The porous CPC particles were prepared by the salt-leaching and molding method [13,15]. X-ray diffraction analysis showed that the crystalline products of the CPC particles were dominated by hydroxyapatite crystal diffraction peaks with low crystallinity (Appendix A). The crystal compositions of the CPC particles were Ca, P and O, which were consistent with the compositions of the inorganic phase of bone tissue (Appendix A). BMP2-CPC was prepared by dropping BMP2 onto CPC particles and evacuating for 30 min to entrap the BMP2 protein in the CPC micropores. The surface topology and microstructure of CPC and BMP2-CPC were observed by scanning electron microscopy (SEM, Hitachi, Japan).

#### 4.2.2. Osteoclast Culture

##### Isolation and Culture of BMMs

BMMs were isolated from 6-week-old C57BL/6 mice and used as precursors for in vitro osteoclast differentiation. Briefly, BMMs were harvested from the bone marrow of the femur and tibia by flushing with Dulbecco’s modified Eagle’s medium (DMEM, Gibco, Carlsbad, CA, USA) containing 10% fetal bovine serum (FBS, Gibco, USA), 1% penicillin/streptomycin (Gibco, USA) and 30 ng/mL M-CSF (R&D, Minneapolis, MN, USA). After 5 days of incubation, the adherent BMMs were detached with a 0.25% trypsin and 0.03% EDTA solution (Gibco, USA) for seeding. The biomaterials were preincubated with DMEM containing 30 ng/mL M-CSF for 2 h prior to seeding. Then, BMMs were seeded at a density of 1 × 10^5^ cells/well into wells containing CPC or BMP2-CPC.

##### Cell Viability of BMMs

The viability of BMMs adhering to CPC or BMP2-CPC was examined on day 1 using a live/dead staining kit (Dojindo, Kumamoto City, Japan). After incubation in phosphate-buffered saline (PBS) containing 2 μM calcein-AM and 1 μM propidium iodide for 15 min, the samples were observed using a laser confocal scanning microscope (Nikon, Tokyo, Japan). The viability of total BMMs was then assessed by the CCK-8 assay (Dojindo, Kumamoto City, Japan) after 1, 3, 5 and 7 days of culture with M-CSF (30 ng/mL). BMMs treated with no biomaterials were set as the control group. The optical density (OD) was measured at a wavelength of 450 nm using a microplate reader (Molecular Devices, San Jose, CA, USA).

##### Tartrate-Resistant Acid Phosphatase (TRAP) Activity

After BMMs were cultured with CPC or BMP2-CPC for 1 day, the culture media were replaced as follows: (1) CPC-M: CPC+BMMs+DMEM containing 30 ng/mL M-CSF; (2) CPC-OC: CPC+BMMs+DMEM containing 30 ng/mL M-CSF and 50 ng/mL RANKL (R&D, USA); (3) BMP2-CPC-M: BMP2-CPC+BMMs+DMEM containing 30 ng/mL M-CSF; and (4) BMP2-CPC-OC: BMP2-CPC+BMMs+DMEM containing 30 ng/mL M-CSF and 50 ng/mL RANKL. On day 7, the cellular TRAP activity was quantified by a TRAP activity assay kit (Beyotime, Shanghai, China) according to the manufacturer’s instructions. The TRAP activity of each sample was determined at a wavelength of 405 nm and then normalized to the corresponding total protein content.

##### Osteoclast Identification

After culturing BMMs with CPC or BMP2-CPC for 1 day, the culture media were replaced as described in the section of TRAP activity. On day 7, the cells adhering to CPC or BMP2-CPC were observed by immunofluorescence staining. The nuclei were stained with DAPI (Beyotime, China), and actin was stained with DyLight™ 594 phalloidin (Cell Signaling Technology, Danvers, MA, USA). Images were acquired with a laser confocal scanning microscope (Nikon, Tokyo, Japan).

##### Calcium Ion Quantification

After culturing BMMs with CPC or BMP2-CPC for 1 day, the culture media were replaced as described in the section of TRAP activity. On day 7, the Ca^2+^ concentration in the culture medium of each group was measured using a Calcium Colorimetric Assay Kit (Beyotime, China) according to the manufacturer’s instructions. The Ca^2+^ concentrations in the collected media were determined by measuring the absorbance at 575 nm and then normalizing it to the Ca^2+^ concentration in pure DMEM.

##### Osteogenic Cytokine Expression and Secretion

After culturing BMMs with CPC or BMP2-CPC for 1 day, the culture media were replaced as described in the section of TRAP activity. On day 7, CPC and BMP2-CPC were removed from the culture media, and the cells adhering to CPC or BMP2-CPC were subjected to total RNA extraction. The PrimeScript RT reagent Kit (Takara, Japan) was used to reverse transcribe the isolated total RNA into cDNA. The mRNA expression levels of collagen triple helix repeat containing 1 (*CTHRC1*), *Wnt10b*, platelet-derived growth factor-BB (*PDGF-BB*) and bone morphogenetic protein 6 (*BMP6*) were quantified by RT-qPCR. The relative expression levels of the target genes were normalized to that of the housekeeping gene *GAPDH*. The primer sequences are listed in Table 4. The PDGF-BB concentration in the culture medium of each group was measured using ELISA kits (Abcam, Cambridge, UK) following the manufacturer’s instructions.

##### Conditioned Medium (CM) Preparation

After culturing BMMs with CPC or BMP2-CPC for 1 day, the culture media were replaced as described In the section of TRAP activity. On day 7, CPC and BMP2-CPC were removed, and the culture medium of each group was collected and centrifuged to obtain the supernatant. The CM was prepared by mixing the supernatants with DMEM containing 10% FBS at a volume ratio of 2:1. The CM groups are illustrated in Table 5.

#### 4.2.3. MSC Culture

##### MSC Isolation and Culture

MSCs were isolated from the femoral and tibial bone marrow of 4-week-old male Sprague–Dawley (SD) rats. Briefly, bone marrow was flushed out from the bone shafts with DMEM, and the cell suspension was centrifuged to remove the supernatants. Then, the cells were seeded in culture dishes and incubated in a humidified atmosphere of 5% CO_2_ at a constant temperature of 37 °C. The culture media were changed every 3 days to remove the nonadherent cells and isolate MSCs. MSCs were passaged when they reached 80–90% confluence using a 0.25% trypsin and 0.03% EDTA solution (Gibco, USA). MSCs were utilized for subsequent experiments at passages 2–4.

##### Cell Viability of MSCs

The viability of MSCs cultured in each CM was assessed by the CCK-8 assay after 3, 7, 10 and 14 days of culture. CCK-8 agent (20 μL) was added to each well and incubated for 2 h at 37 °C. The OD was measured at a wavelength of 450 nm using a microplate reader.

##### Alkaline Phosphatase (ALP) Activity and Alizarin Red S (ARS) Staining

MSCs were cultured in the CMs mentioned in the section of CM Preparation. After 5 and 7 days of culture, ALP staining was performed with the BCIP/NBT Alkaline Phosphatase Color Development Kit (Beyotime, Hangzhou, China) in accordance with the manufacturer’s instructions. ALP activity was quantified using an ALP activity assay kit (Beyotime, China) according to the manufacturer’s instructions. The relative ALP activity was measured at a wavelength of 405 nm and then normalized to the corresponding total protein content. After 14 and 21 days of culture, MSCs were fixed with 10% formalin and then stained with ARS. To quantify MSC mineralization, the precipitated ARS was dissolved in 10% cetylpyridinium chloride for absorbance measurement. The relative calcium deposition was measured at a wavelength of 562 nm and then normalized to the corresponding total protein content.

##### Osteogenic Differentiation of MSCs

MSCs were cultured in the CMs mentioned in the section of CM Preparation. After 5 days of culture, immunofluorescence staining was performed to determine the expression of collagen-I (COL-I) in the MSCs of each group. MSCs were fixed with 4% PFA, permeabilized in 0.1% Triton X-100 (Beyotime, China) and blocked with bovine serum albumin (BSA, Gibco, USA) before being incubated with a primary antibody against COL-I (Abcam, Cambridge, UK) overnight at 4 °C. The MSCs were then incubated with an Alexa Fluor^®^ 488-conjugated goat anti-rabbit IgG antibody (Abcam, UK) for 1 h at room temperature. The nuclei were stained with DAPI (Beyotime, China), and actin was stained with DyLight™ 594 phalloidin (Cell Signaling Technology, Danvers, MA, USA). Images were acquired with a fluorescence microscope (Nikon, Japan).

After 12 days of culture, Western blotting was performed to determine the expression of Runt-related transcription factor 2 (Runx2), osteopontin (OPN) and OCN in the MSCs of each group. MSCs were lysed in RIPA lysis buffer (Beyotime, China) containing 1 mM phenyl-methyl sulfonylfluoride (PMSF, Beyotime, China) to extract total protein. The total protein concentrations were measured using a BCA assay kit (Beyotime, China). Proteins were separated by sodium dodecyl sulfate–polyacrylamide gel electrophoresis (SDS-PAGE) and then transferred onto polyvinylidene difluoride (PVDF) membranes (Millipore, USA). After blocking with skim milk at room temperature for 1 h, the membranes were incubated with primary antibodies against Runx2 (Abcam, UK), OPN (Abcam, UK), OCN (Abcam, UK) and β-actin (Proteintech, Chicago, IL, USA) at 4 °C overnight. After three washes with PBS-Tween buffer, the membranes were incubated with HRP-conjugated secondary antibodies for 1 h at room temperature. The protein bands were visualized using an Odyssey infrared imaging system (LI-COR, Lincoln, NE, USA). The relative intensities of the protein bands were quantified by ImageJ.

### 4.3. In Vivo Study

#### 4.3.1. Study Design

The animal experiments were performed in accordance with the National Institutes of Health guide for the care and use of laboratory animals (NIH Publications No. 8023, revised 1978). Eighteen 3-month-old male New Zealand white rabbits were acquired from the Animal Center of Shanghai Ninth People’s Hospital and maintained at a temperature of 22 °C on a 12-h light/12-h dark cycle with free access to water and food. A total of 36 calvarial defects (10 mm diameter) were randomly divided into 3 groups as follows: (1) Con (unfilled defects); (2) CPC (defects filled with CPC); and (3) BMP2-CPC (defects filled with BMP2-CPC). The present study complied with the principles of laboratory animal care and was authorized by the Animal Research Ethics Committee of Shanghai Ninth People’s Hospital affiliated with Shanghai Jiao Tong University, School of Medicine. Six rabbits were euthanized at 1.5 months after surgery, and the rest were euthanized at 3 months after surgery. The obtained calvarial samples were fixed in 4% PFA.

#### 4.3.2. Surgical Procedures

After general anesthetization with a mixture of ketamine 35 mg/kg and xylazine 5 mg/kg, the dorsal part of the scalp covering the calvarias was shaved and disinfected with iodophor gauze. A dermoperiosteal incision was made along the midline of the cranium from the frontal bone to the occipital bone. A full-thickness flap was elevated to expose the surgical area. Two 10-mm-diameter calvarial defects were created on both sides of the midline using a trephine bur under profuse saline irrigation. The resected bone blocks were carefully removed, and the defects were treated as described in Section 4.3.1. Then, the periosteum and scalp skin were sutured in layers with polylactic acid sutures (Vycril 5.0, Ethicon, Raritan, NJ, USA) and nylon monofilament sutures (ETHILON^®^ Nylon Suture, Ethicon, Raritan, NJ, USA). The animals were intramuscularly injected with antibiotics two times daily for 3 days.

#### 4.3.3. Microcomputed Tomography (micro-CT) Examination

Micro-CT scanning (Bruker Skyscan, Aartselaar, Belgium) was used to evaluate bone regeneration within the defect area. Scanning was performed at 100 kV and 100 μA with a thickness of 26 μm per slice. To analyze the bone regeneration within the defect area, a central 9-mm-diameter region within the 10-mm-diameter defect area was defined as the region of interest (ROI) to exclude the native bone margins. The newly formed bone in the ROI was analyzed by determining the bone volume/total volume (BV/TV) and bone mineral density (BMD).

#### 4.3.4. Histological Examination (Undecalcified)

Half of the 3-month samples were randomly selected and embedded without decalcification in methyl methacrylate (MMA) blocks (Merck, Darmstadt, Germany) after a series of dehydration and tissue-clearing steps. After block trimming and grinding, the blocks were sliced from the centers of the samples using an Exakt diamond cutter (Kulzer Exakt, Wehrheim, Germany). The new bone formation and degradation of material particles were evaluated via the HE staining of undecalcified samples. The relative areas of newly formed bone and residual biomaterial particles were measured in five randomly selected fields of vision using ImageJ.

#### 4.3.5. Histological and Immunohistochemical Examinations (Decalcified)

The 1.5-month samples and the remaining 3-month samples were decalcified in a 10% EDTA solution. Following dehydration in a series of gradient ethanol (70–100%), the decalcified samples were embedded in paraffin for sectioning. The samples were sectioned from the center of the defects, following the coronal plane of the calvarial bone. Sections 5-μm thick were stained with HE to assess bone regeneration in the defect area. The relative area of newly formed bone was measured in five randomly selected fields of vision using ImageJ.

Immunohistochemical staining of the decalcified sections was performed to evaluate the expression of OPN and OCN. Following deparaffinization and rehydration, the endogenous peroxidase activity in the prepared sections was blocked with 3% H_2_O_2_. Then, the sections were subjected to antigen retrieval prior to incubation with primary antibodies against OPN (Novus Biologicals, Littleton, CO, USA) and OCN (Abcam, UK). After incubation at 4 °C overnight, the sections were treated with biotinylated secondary antibodies and incubated with streptavidin–horseradish peroxidase complex (Thermo Fisher Scientific, Waltham, MA, USA). Subsequently, the sections were incubated with 3,3-diaminobenzidine (DAB) and counterstained with hematoxylin. The stained sections were captured under an upright microscope (Nikon, Tokyo, Japan).

#### 4.3.6. TRAP Staining

After deparaffinization and rehydration, the decalcified sections were stained with a TRAP staining kit (Servicebio, Wuhan, China) according to the manufacturer’s instructions. All sections were counterstained with hematoxylin. The TRAP-positive areas manifested as red, and the relative TRAP-positive areas were measured in five randomly selected fields of vision using ImageJ.

### 4.4. Statistical Analysis

Numerical data are presented as the mean ± standard deviation (SD), and statistical analysis was performed using SPSS 22.0 (IBM, Armonk, NY, USA). For the in vitro and in vivo studies, an unpaired *t*-test was performed to identify differences between two groups. Multigroup data were analyzed by one-way or two-way analysis of variance (ANOVA) followed by Tukey’s post-hoc test. The clinical data were statistically analyzed by the Mann–Whitney test. The levels of significance were taken as * *p* < 0.05 or ** *p* < 0.01.

## 5. Conclusions

In conclusion, BMP2-CPC facilitates a harmonized pattern of bone regeneration, bone remodeling and biomaterial resorption through the induction of osteoclastogenesis (Figure 7c). As depicted in Figure 7d, BMP2 released from BMP2-CPC enhanced the osteoclast differentiation of BMMs induced by M-CSF and RANKL. The increased osteoclasts promoted the osteogenic differentiation of MSCs by secreting calcium ions, CTHRC1 and PDGF-BB, thereby enhancing local bone regeneration. With the accumulation of new bone, osteoclasts participated in the bone remodeling process to propel the maturity of the bone structure at defect sites. Meanwhile, the increased osteoclasts accelerated the in vivo resorption of BMP2-CPC, identifying BMP2-CPC as a better choice than CPC and DBB for repairing alveolar bone defects. These in vitro and in vivo results were highly consistent with the clinical results. Taken together, these results highlight the promise of BMP2-CPC for the treatment of alveolar bone defects not only due to its enhanced osteogenic performance but also its optimized properties in osteoclast activation.

## Figures and Tables

**Figure 1 ijms-23-12204-f001:**
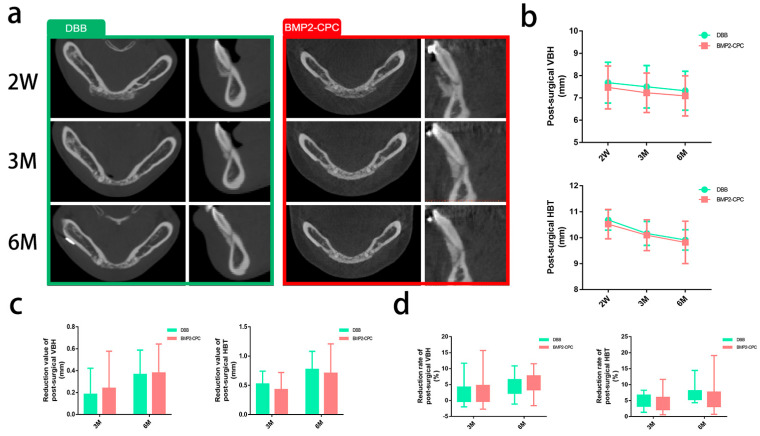
**Clinical radiographic evaluation results.** (**a**) Representative postsurgical CT images of each group; (**b**) measurement of the postsurgical VBH and HBT values; (**c**) reduction values of the postsurgical VBH and HBT of each group; (**d**) reduction rates of the postsurgical VBH and HBT of each group.

**Figure 2 ijms-23-12204-f002:**
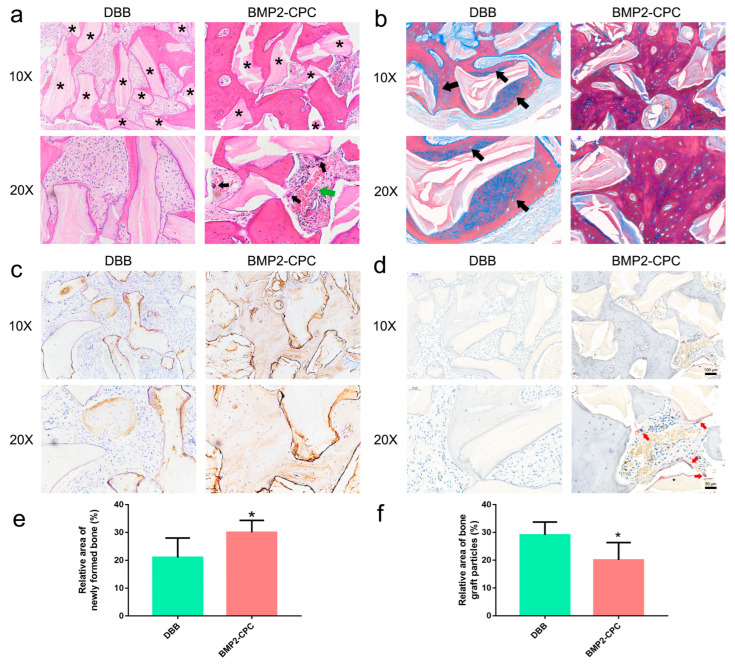
**Clinical histological examination results.** (**a**) HE staining results (* denotes residual bone graft particles, black arrows mark multinucleated osteoclast cells, green arrow marks vascularized bone marrow-like tissues); (**b**) Masson staining results (black arrows mark the stained immature bone); (**c**) immunohistochemical staining results for OCN; (**d**) TRAP staining results (red arrows mark the stained multinucleated osteoclasts); (**e**) quantification of the area of newly formed bone in each group; (**f**) quantification of the area of residual bone graft particles in each group.

**Figure 3 ijms-23-12204-f003:**
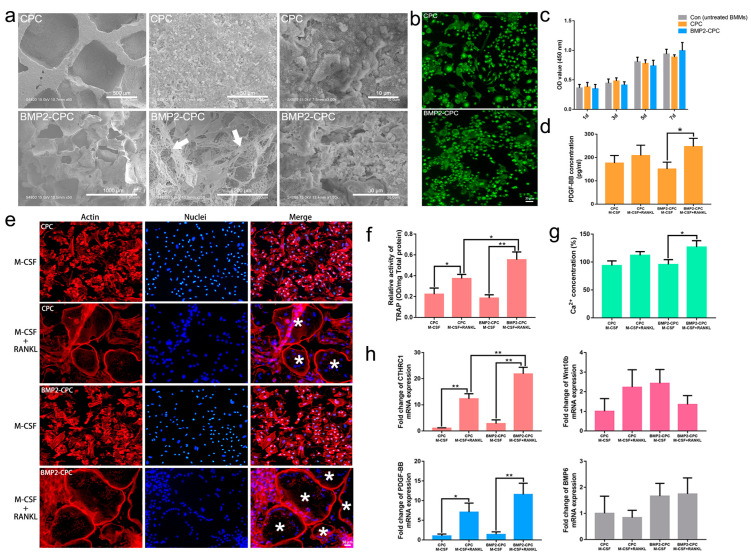
**Biomaterial microstructure and osteoclast differentiation.** (**a**) SEM observation of the biomaterial microstructure (white arrows mark the flocculent BMP2 protein); (**b**) viability of attached BMMs; (**c**) viability of total BMMs of each group; (**d**) ELISA examination of PDGF-BB secretion; (**e**) identification of osteoclast differentiation (* denotes multinucleated osteoclasts); (**f**) relative activity of TRAP in each group; (**g**) relative concentration of calcium ions in each group; (**h**) PCR analysis of the gene expression levels of osteoclast-secreted osteogenic factors (*CTHRC1*, *PDGF-BB*, *Wnt10b*, *BMP6*) in each group.

**Figure 4 ijms-23-12204-f004:**
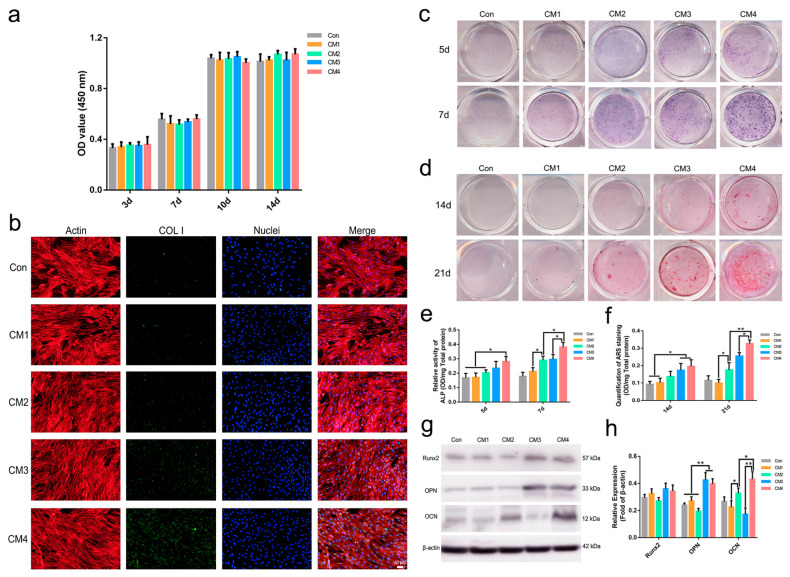
**MSC activity and differentiation.** (**a**) Viability of MSCs in each group; (**b**) immunofluorescence staining of COL-I in MSCs cultured in various CMs; (**c**) ALP staining of MSCs cultured in various CMs; (**d**) ARS staining of MSCs cultured in various CMs; (**e**) relative ALP activity of MSCs cultured in various CMs; (**f**) quantification of the ARS staining of MSCs cultured in various CMs; (**g**) Western blotting results for the expression of Runx2, OPN and OCN in MSCs cultured in various CMs; (**h**) quantification of the Western blotting results.

**Figure 5 ijms-23-12204-f005:**
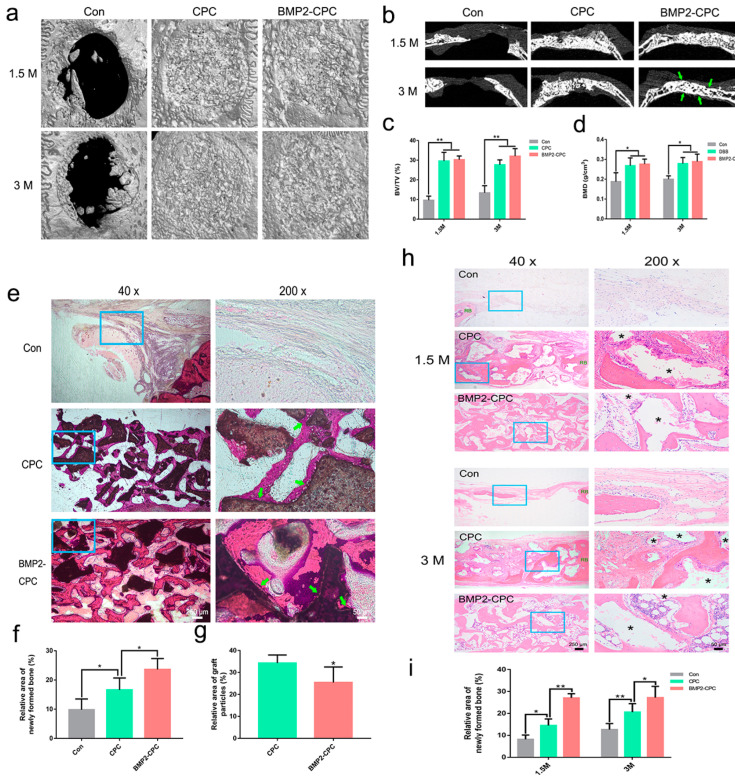
**Micro-CT analysis and HE staining results.** (**a**) Three-dimensional reconstruction images of the defects in each group; (**b**) transverse micro-CT images of the defects in each group (green arrows mark the two-layer cortical bone-like structure); (**c**) quantitative analysis of the proportion of new bone volume in the defects in each group; (**d**) quantitative analysis of the bone mineral density in the defects in each group; (**e**) HE-stained images of the in vivo healing pattern in the defects in each group (undecalcified, 3 months after the surgery, green arrow marks the close contact between newly formed bone and graft particles); (**f**) quantification of the area of newly formed bone in the defects in each group; (**g**) quantification of the area of graft particles in the defects in each group; (**h**) HE-stained images of the in vivo healing pattern in the defects in each group (decalcified, * denotes residual graft particles); (**i**) quantification of the area of newly formed bone in the defects in each group.

**Figure 6 ijms-23-12204-f006:**
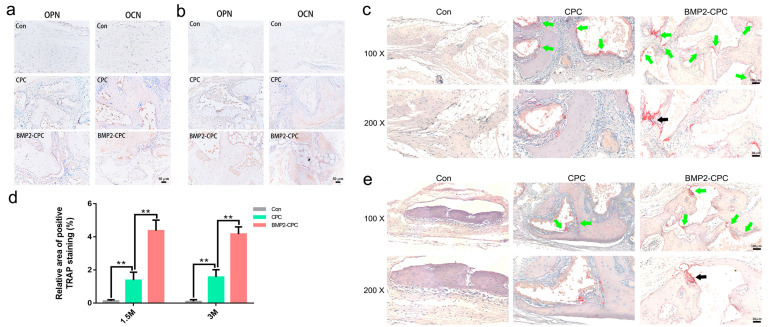
**Immunohistochemical staining and TRAP staining results.** (**a**) Representative immunohistochemical staining images of OPN and OCN in each group (200×, 1.5 months after surgery); (**b**) representative immunohistochemical staining images of OPN and OCN in each group (200×, 3 months after surgery); (**c**) 1.5-month TRAP staining results of each group; (**d**) quantification of the area of positive TRAP staining in the defects in each group; (**e**) 3-month TRAP staining results of each group.

**Figure 7 ijms-23-12204-f007:**
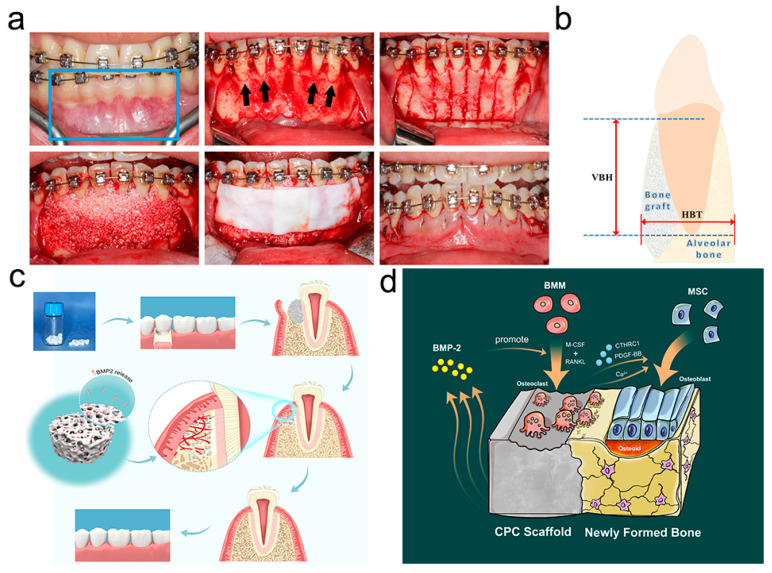
**PAOO surgery and Schematic diagrams.** (**a**) PAOO surgery in the anterior mandibular region (blue box marks the surgical region, and black arrows mark the sites of alveolar bone dehiscence); (**b**) schematic diagram of the radiographic evaluation; (**c**) schematic diagram of BMP2-CPC-induced alveolar bone regeneration; (**d**) schematic diagram of osteoclast-mediated osteogenesis, bone remodeling and biomaterial resorption.

**Table 1 ijms-23-12204-t001:** Measurement results.

Parameters	DBB (mm)	BMP2-CPC (mm)	*p* Value
**VBH_2W_**	7.68 ± 0.89	7.47 ± 0.94	0.663
**VBH_3M_**	7.50 ± 0.92	7.23 ± 0.86	0.258
**VBH_6M_**	7.32 ± 0.85	7.09 ± 0.88	0.358
**HBT_2W_**	10.69 ± 0.38	10.52 ± 0.55	0.358
**HBT _3M_**	10.16 ± 0.45	10.10 ± 0.58	0.893
**HBT_6M_**	9.91 ± 0.38	9.82 ± 0.79	0.869

**Table 2 ijms-23-12204-t002:** Analysis of VBH and HBT.

Parameters	DBB	BMP2-CPC	*p* Value
**3-month reduction value of VBH (mm)**	0.18 ± 0.23	0.24 ± 0.33	0.685
**6-month reduction value of VBH (mm)**	0.36 ± 0.22	0.38 ± 0.26	0.940
**3-month reduction rate of VBH (%)**	2.45 ± 3.25	3.01 ± 4.11	0.663
**6-month reduction rate of VBH (%)**	4.71 ± 2.88	4.98 ± 3.12	0.988
**3-month reduction value of HBT (mm)**	0.52 ± 0.21	0.43 ± 0.28	0.142
**6-month reduction value of HBT (mm)**	0.77 ± 0.30	0.70 ± 0.49	0.258
**3-month reduction rate of HBT (%)**	4.90 ± 2.01	4.05 ± 2.64	0.189
**6-month reduction rate of HBT (%)**	7.21 ± 2.78	6.76 ± 4.87	0.284

**Table 3 ijms-23-12204-t003:** Analysis formulas.

	Vertical Bone Height	Horizontal Bone Thickness
**Reduction value-3M**	VBH_2W_ − VBH_3M_	HBT_2W_ − HBT_3M_
**Reduction rate-3M**	(VBH_2W_ − VBH_3M_)/VBH_2W_ × 100%	(HBT_2W_ − HBT_3M_)/HBT_2W_ × 100%
**Reduction value-6M**	VBH_2W_ − VBH_6M_	HBT_2W_ − HBT_6M_
**Reduction rate-6M**	(VBH_2W_ − VBH_6M_)/VBH_2W_ × 100%	(HBT_2W_ − HBT_6M_)/HBT_2W_ × 100%

VBH_2W_: 2-week vertical bone height; VBH_3M_: 3-month vertical bone height; VBH_6M_: 6-month vertical bone height; HBT_2W_: 2-week horizontal bone thickness; HBT_3M_: 3-month horizontal bone thickness; HBT_6M_: 6-month horizontal bone thickness.

**Table 4 ijms-23-12204-t004:** PCR primer pairs (BMM-derived cells).

Gene	Primer Sequences
** *CTHRC1* **	Forward: 5′-AAGCAAAAAGCGCTGATCC-3′
	Reverse: 5′-CCTGCTGGTCCTTGTAGACAC-3′
** *Wnt10b* **	Forward: 5′-GGGCTCAGGTTCCTACTTCC-3′
	Reverse: 5′-AAGGAGAAGCCTCCCAAGAG-3′
** *PDGF-BB* **	Forward: 5′-CCTCGGCCTGTGACTAGAAG-3′
	Reverse: 5′-CCTTGTCATGGGTGTGCTTA-3′
** *BMP6* **	Forward: 5′-AACCTTTCTTATCAGCATTTACCA-3′
	Reverse: 5′-GTGTCCAACAAAAATAGGTCAGAG-3′
** *GAPDH* **	Forward: 5′-TGGTGAAGGTCGGTGTGAAC-3′
	Reverse: 5′-CCATGTAGTTGAGGTCAATGAAGG-3′

**Table 5 ijms-23-12204-t005:** The groups of CM.

Group	Material	M-CSF	RANKL	Corresponding Cells	DMEM Containing 10% FBS
**Con**	—	—	—	None	+
**CM1**	CPC	+	—	BMMs	+
**CM2**	CPC	+	+	Osteoclasts	+
**CM3**	BMP2-CPC	+	—	BMMs	+
**CM4**	BMP2-CPC	+	+	Osteoclasts	+

“+” added, “—” not added.

## Data Availability

The datasets used and analyzed during the current study are available from the corresponding author on reasonable request.

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
