# Peer review of "Osteoclast-Driven Osteogenesis, Bone Remodeling and Biomaterial Resorption: A New Profile of BMP2-CPC-Induced Alveolar Bone Regeneration"

_ijms, 2022, doi:10.3390/ijms232012204_

Round 1
Reviewer 1 Report
In this study, the authors demonstrated that a BMP2-CPC scaffold can enhance bone regeneration and osteogenesis via BMP2-mediated stimulation of osteoclastogenesis, which led to increased secretion of the pro-osteogenic factors PDGF-BB and CTHRC1, as well as elevated Ca2+ levels within the culture medium. Overall, the manuscript is generally quite well-written, with sound experimental methodology and comprehensive results that would certainly be of interest to researchers in the biomaterials and bone tissue engineering fields. Nevertheless, the authors should address the following points before the manuscript is acceptable for publication.
(1) Introduction - besides BMP2, there are also other pro-osteogenic growth factors that have been utilized in bone tissue engineering. Why was BMP2 specifically chosen? Is BMP2 superior to other osteoinductive factors for bone tissue engineering applications?
(2) Introduction - should mention deficiency of using protein-based growth factors, for example short half-life and easily degraded in vivo. How about controlled release of pro-osteogenic small molecules from CPC scaffolds?
(3) Introduction - why did the authors chose CPC over β-TCP and HA? What are the advantages of CPC over β-TCP and HA?
(4) Materials and methods - why is there discrepancy in experimental groups between the clinical, animal and in vitro studies? For example, there is no CPC control group in the clinical study. Likewise, there is no DBB group in the animal and in vitro studies?
(5) Results - Fig 3 - besides cell viability assay, a CCK8-based proliferation assay would be good.
(6) Results - Fig 3 - besides qRT-PCR analysis of mRNA expression, it would be good to have ELISA-based assays to directly quantify secretory levels of PDGF-BB and CTHRC1 within the conditioned media.
(7) Discussion - besides acid-based resorption of CPC, are there also enzymes secreted by osteoclasts that contribute to degradation of CPC?
(8) Discussion - explain the discrepancy between the radiographic results presented in fig 1 versus histology results presented in fig 2. The radiographic results showed no significant differences, whereas there are significant differences between groups in the histology results?
Author Response
RE: ijms-1924395, entitled " Osteoclast-driven Osteogenesis, Bone Remodeling and Biomaterial Resorption: A New Profile of BMP2-CPC-induced Alveolar Bone Regeneration".
We thank the reviewer for his careful read and thoughtful comments on previous draft. We have carefully taken his comments into consideration in preparing our revision, which has resulted in a paper that is clearer, more compelling, and broader. The following summarizes how we respond to the reviewer’s comments.
Suggestion:
In this study, the authors demonstrated that a BMP2-CPC scaffold can enhance bone regeneration and osteogenesis via BMP2-mediated stimulation of osteoclastogenesis, which led to increased secretion of the pro-osteogenic factors PDGF-BB and CTHRC1, as well as elevated Ca2+ levels within the culture medium. Overall, the manuscript is generally quite well-written, with sound experimental methodology and comprehensive results that would certainly be of interest to researchers in the biomaterials and bone tissue engineering fields. Nevertheless, the authors should address the following points before the manuscript is acceptable for publication.
(1) Introduction - besides BMP2, there are also other pro-osteogenic growth factors that have been utilized in bone tissue engineering. Why was BMP2 specifically chosen? Is BMP2 superior to other osteoinductive factors for bone tissue engineering applications?
(2) Introduction - should mention deficiency of using protein-based growth factors, for example short half-life and easily degraded in vivo. How about controlled release of pro-osteogenic small molecules from CPC scaffolds?
(3) Introduction - why did the authors chose CPC over β-TCP and HA? What are the advantages of CPC over β-TCP and HA?
(4) Materials and methods - why is there discrepancy in experimental groups between the clinical, animal and in vitro studies? For example, there is no CPC control group in the clinical study. Likewise, there is no DBB group in the animal and in vitro studies?
(5) Results - Fig 3 - besides cell viability assay, a CCK8-based proliferation assay would be good.
(6) Results - Fig 3 - besides qRT-PCR analysis of mRNA expression, it would be good to have ELISA-based assays to directly quantify secretory levels of PDGF-BB and CTHRC1 within the conditioned media.
(7) Discussion - besides acid-based resorption of CPC, are there also enzymes secreted by osteoclasts that contribute to degradation of CPC?
(8) Discussion - explain the discrepancy between the radiographic results presented in fig 1 versus histology results presented in fig 2. The radiographic results showed no significant differences, whereas there are significant differences between groups in the histology results?
Response:
- We added related descriptions to Introduction according to the reviewer’s suggestion. “…, various growth factors (bone morphogenetic protein 2, transforming growth factor-β, insulin-like growth factor I, etc.) have been introduced into synthetic bone substitutes to improve their osteoinductivity. Among these growth factors, bone morphogenetic protein 2 (BMP2) is thought to be the most potent osteoinductive factor and has been widely reported to significantly promote the osteogenic differentiation of MSCs and new bone formation in both experimental and clinical practice.”
- We added related descriptions to Introduction according to the reviewer’s suggestion. “However, due to the short half-life of BMP2, the in vivo BMP2 is prone to degradation, resulting in that the local effective concentration of BMP2 is hard to be maintained. To this end, controlled-release scaffolds are used to preserve the activity of BMP2. Lin et al. indicated that the present BMP2-CPC biomaterial showed an initial burst release within 24 hours followed by a slow sustained release. The structural analysis results demonstrated that the secondary structure of CPC-released BMP2 was well maintained, and the sustained release effectively established local accumulation of BMP2.”
- We added related descriptions to Introduction according to the reviewer’s suggestion. “Compared to the over slow resorption rate of HA and the over fast resorption rate of β-TCP, the improved resorption rate of CPC is more suitable for guiding bone regeneration.”
- In order to achieve a satisfactory bone healing outcome, BMP2-CPC was a better choice than CPC for clinical study due to its enhanced osteogenic effect. DBB was the currently frequently-used biomaterial for PAOO surgery, and, therefore, DBB was selected as the control group for clinical study. Through evaluating the clinical results, we found that compared to DBB, BMP2-CPC was capable of significantly enhancing osteocalstogenesis. According to the results of our pre-experiments (Fig.1), both DBB and CPC group had no significant influence on TRAP activity compared to the control group, which meant that DBB and CPC had no significant influence on osteoclastogenesis. Thus, we set CPC as the control group and put emphasis on investigating the mechanism of BMP2-CPC-stimulated osteoclastogenesis in the following in vitro and in vivo studies. Through comparing BMP2-CPC with CPC, the BMP2-CPC-induced pro-osteoclastogenesis effect was subsequently revealed by the experimental studies.
Fig.1 TRAP activity
- We added CCK-8 results to Fig.3 according to the reviewer’s suggestion.
- We added the ELISA results of PDGF-BB secretion to Fig.3 according to the reviewer’s suggestion. However, due to the lack of commercial products of ELISA kit for mouse CTHRC1, we are unable to present the results regarding to the secretory level of CTHRC1.
- CPCs are generally degraded through two main mechanisms, passive dissolution and active degradation through cellular activity. For osteoclasts, they contribute to CPC degradation through an acidic mechanism to reduce the micro-environmental pH which results in demineralization of CPC[1, 2]. The enzymes secreted by osteoclasts, such as Cathepsin K and matrix metalloproteinase 9, mainly contribute to the resorption of the organic component of bone[3, 4]. Thus, considering that CPC is an inorganic biomaterial, the acid-based resorption mechanism might dominate the active degradation via osteoclasts.
- We added related descriptions explaining the discrepancy between the radiographic results presented in Fig 1 versus histological results presented in Fig 2. “However, the findings derived from the clinical radiographic examination should be interpreted with caution due to the similar inorganic compositions of DBB and BMP2-CPC to those of bone tissues that might interfere the identification of bone area in CT images. Therefore, histological examinations were performed to further reveal the bone regeneration pattern induced by DBB or BMP2-CPC.”
- Schröter, L.; Kaiser, F.; Stein, S.; Gbureck, U.; Ignatius, A., Biological and mechanical performance and degradation characteristics of calcium phosphate cements in large animals and humans. Acta Biomater 2020, 117, 1-20.
- Zeeshan, S.; Mohamed-Nur, A.; Ahmed, H.; Syed, M.; Haroon, R.; Michael, G., Mechanisms of in Vivo Degradation and Resorption of Calcium Phosphate Based Biomaterials. Materials (Basel) 2015, 8, (11), 7913-7925.
- Kim, J. M.; Lin, C.; Stavre, Z.; Greenblatt, M. B.; Shim, J. H., Osteoblast-Osteoclast Communication and Bone Homeostasis. Cells 2020, 9, (9).
- Kylmaoja, E.; Nakamura, M.; Tuukkanen, J., Osteoclasts and Remodeling Based Bone Formation. Current stem cell research & therapy 2016, 11, (8), 626-633.

Reviewer 2 Report
Congratulations! I've recommended the article be published.
These are a few notes about the article Osteoclast-driven Osteogenesis, Bone Remodeling, and Bio-2 material Resorption: A New Profile of BMP2-CPC-induced Alveolar Bone Regeneration. My opinion is that the article is suitable for publication.
The paper highlights the role of BMP2-CPC in bone restoration by materials’ excellent biocompatibility, osteoconductivity, osteoinductivity, and immunoregulatory properties. The clinical applicability of the studies conducted by the authors was demonstrated in vivo and in vitro. The practical impact of their findings is great and sound as the scientific impact.
The paper clarifies the effect of BMP2-CPC on osteoclastogenesis and subsequent osteoclast-driven osteogenesis, bone remodeling, and biomaterial resorption.
Results were sustained by good statistics, and photos that show bone regeneration.
I don’t think methods should be more detailed because there are well-known by the scientific community.
In vitro and in vivo results were highly consistent with the clinical results.
The article is well-written and very good from my point of view.
Author Response
RE: ijms-1924395, entitled " Osteoclast-driven Osteogenesis, Bone Remodeling and Biomaterial Resorption: A New Profile of BMP2-CPC-induced Alveolar Bone Regeneration".
Reviewer comment
Congratulations! I've recommended the article be published.
These are a few notes about the article Osteoclast-driven Osteogenesis, Bone Remodeling, and Bio-2 material Resorption: A New Profile of BMP2-CPC-induced Alveolar Bone Regeneration. My opinion is that the article is suitable for publication.
The paper highlights the role of BMP2-CPC in bone restoration by materials’ excellent biocompatibility, osteoconductivity, osteoinductivity, and immunoregulatory properties. The clinical applicability of the studies conducted by the authors was demonstrated in vivo and in vitro. The practical impact of their findings is great and sound as the scientific impact.
The paper clarifies the effect of BMP2-CPC on osteoclastogenesis and subsequent osteoclast-driven osteogenesis, bone remodeling, and biomaterial resorption.
Results were sustained by good statistics, and photos that show bone regeneration.
I don’t think methods should be more detailed because there are well-known by the scientific community.
In vitro and in vivo results were highly consistent with the clinical results.
The article is well-written and very good from my point of view.
Response
We thank the reviewer for his careful read and scrutinizing. We deeply appreciate for the reviewer’s high comment!
Round 2
Reviewer 1 Report
The authors have addressed my comments. This manuscript could be accepted.